# Planning of Safety of Cities and Territory from the Point of View of Population Protection in the Czech Republic

**Jiri Pokorny** [1],*, **Barbora Machalova** [1], **Simona Slivkova** [1], **Lenka Brumarova** [1] and **Vladimir Vlcek** [2]

1   Faculty of Safety Engineering, VSB—Technical University of Ostrava, Lumirova 13/630,
    700 30 Ostrava-Vyskovice, Czech Republic; barbora.machalova@vsb.cz (B.M.); simona.slivkova@vsb.cz (S.S.);
    lenka.brumarova@vsb.cz (L.B.)
2   Fire Rescue Services of the Moravian-Silesian Region, Vyskovicka 40, 700 30 Ostrava-Zabreh,
    Czech Republic; vladimir.vlcek@hzsmsk.cz
*   Correspondence: jiri.pokorny@vsb.cz

**Abstract:** Ensuring territorial safety is one of the state's main tasks, and the public administration plays a primary role in fulfilling it. The safety and sustainability of a territory is ensured by, inter alia, safety planning, including spatial planning. Spatial planning contains safety requirements for population protection. The requirements for ensuring the safety of the population included in the spatial plan vary significantly between countries. The existing population protection requirements contained in the spatial plan have been unsatisfactory for a long time. The main issue is that they are out of date and difficult to apply. The article presents a new proposed method for determining requirements for the protection of the population, which is based on evaluating the risks in the cities and territory. The specific population protection requirements are determined based on the resulting risks and their scaling. The requirements are classified as general when the territory is not faced with external risks or specific when there are such risks. The method is applied to the conditions of the Czech Republic. In terms of national standard specifics, there are requirements in areas of public infrastructure, public utility buildings, and public benefit measures. The method for determining population protection requirements can be considered applicable in a general form by various countries if the national standards specifics or other aspects are taken into account.

**Keywords:** sustainability of the territory; population protection; civil protection; territorial risks; safety planning

## 1. Introduction

Territorial safety and population protection is one of the basic tasks of the state and one of the primary interests of the European Union too. It is conceived as its safety policy. The principles of the European Union's safety policy include, for example, increasing the citizens' prosperity, ensuring sustainable territorial development, and promoting the resilience of democratic systems [1].

Among the European Union protection documents, we can find the Action Program, Living well within the limits of our planet, which in particular deals with the climate change, natural and biological diversity, the environment, human health and quality of life, natural resources and their sustainability, and waste [2].

The territorial safety is closely linked to long-term sustainability. Sustainability strategies are associated with the issues of "people and society", "economic model", "resilient ecosystems", "municipalities and regions", "global development", and "good governance" [3,4].

The sustainability of a territory is closely related to the preparedness to deal with extraordinary incidents, crisis situations, and disasters. For their solution within the European Union, an integrated approach known as the civil protection mechanism is necessary. The integrated approach is primarily focused on the protection of people, the environment, and property, including cultural heritage, against all types of disasters, including the response to them. An important position here is that of prevention, which refers to any measure focused on the reduction of risk or mitigation of the adverse effects of disasters. Within prevention comes risk mapping and assessment, which is carried out by the Member States of the European Union [5,6].

The ways and principles of the risk assessment have a historical background based on the so-called "Yokohama Strategy for a Safer World", which included the fundamental principles for a reduction of naturogenic disaster risks, mitigation of their consequences, and adaptations [7]. The principles for the disaster reduction in 2005 in Hyogo further directed the development of sending the identification of risks and the investment in disaster preparedness. The main direction was set as building the resilience of nations and communities against disasters [8]. An evaluation of previous plans for disaster risk reduction and the identification of further solution procedures in the field of risk assessment came in 2015 in Sendai. The basis for the implementation of preventive measures is the survey of the nature the risks. At the same time, it turned out that for the disaster solution, it is essential to ensure the preparedness of authorities at all levels [9,10].

The crucial part of the European Union's civil protection mechanisms is not only the risk assessment, but also the sharing of knowledge, best practices, and information. The Member States of the European Union carry out a risk assessment every three years. The importance of risk prevention is emphasized.

Risk assessment in the Czech Republic was carried out at the national level in 2015. The identified threats to the Czech Republic were one of the foundations for the elaboration of a new concept of ensuring the security of a territory from the point of view of population protection. Based on the risk assessment, the Member States of the European Union are required to improve the procedures of disaster management planning at the national or equivalent lower level. The newly proposed procedure responds to this requirement.

Territorial safety and population safety in the Czech Republic are also related to the protection of the environment. It is included in the state's environmental policy. Part of the policy is related to environmental protection and protection against floods, drought, fire, slope instabilities, and places of contamination [11].

The principles for ensuring the safety of a territory is addressed in safety planning. Emergency plans, contingency plans, and related plans are among the most important "planning documents".

The preparation of safety plans is closely associated with spatial planning, which in the Czech Republic is divided into spatial analysis documents and spatial planning documentation [12–14]. The tasks of spatial planning include assessing the state of the territory, its natural, cultural, and civilizational values, and setting requirements for the use and arrangement of the territory, buildings, and public space. All territorial requirements are addressed with regard to ensuring the safety of the territory. The principles are set out in the territorial development policy [15].

The requirements for spatial planning abroad are also related to ensuring the safety and sustainability of the territory. Examples of countries that have a somewhat similar safety system to the Czech Republic include Switzerland and Germany. Spatial planning in Switzerland is regulated by the Act on Spatial Planning [16]. Spatial planning includes requirements for the use and arrangement of the territory, protection of natural and cultural values, creation of conditions for economic development, sustainability of the territory, and ensuring the protection of the population. In Germany, spatial planning consists of the development and organization of the territory, the pursuit of balanced social, cultural, economic, and environmental conditions, and the consideration of defense and civil protection requirements [17].

In the risk assessment of the territory, it seems to be purposeful to take into account only the most significant selected hazards that have a major impact on the environment, cultural, and historical values of the territory, the territorial development, and the landscape creation in accordance with the principles of the long-term sustainable development. It is therefore appropriate to use a qualitative assessment of the selected hazards that allow to specify the most significant risks. The selected risks have been also taken into account in the spatial planning [18].

In the risk assessment of the large areas, the technique of multidimensional analysis has been used. The results of the analysis enable to better identify the risk areas from the perspective of naturogenic and anthropogenic risks, to classify them, and to take them into account in the planning. In the final consequence, it is possible to allocate the sufficient forces and resources to protect the territory [19].

The risk identification must be also accompanied by a quantification process. However, in some cases, the quantification of risks in relation to the possible occurrence and development of disasters has not been selected optimally. This state causes the need to develop new techniques and procedures for assessing the risks of the territory [20].

For the assessment of the environmental safety in the territory, the system of selected specific indicators that can identify the level of safety has usually been used [21]. The use of indicators requires, in some cases, modifications in relation to the solved territory [22].

The assessment of risks in the territory in connection with ensuring the sustainability can also be performed in relation to military conflicts. It is possible to determine a complex level of the risk with the division of the territory into subcategories. To process the results, it is possible to use the standard geographic systems, which can be beneficial for improving the environmental management. An example of application is the conflict in the eastern part of Ukraine [23].

In terms of the territorial security, it is possible to speak about the assessment of "territorial resilience" as a concept of helping with the identification of vulnerabilities and with the establishment of the preventive measures in the form of security plans [24].

The present has been characterized by a complex assessment of territorial risks, i.e., by the assessment of "multi-vulnerability", "multi-exposure", and "multi-preparedness". It is obvious that the issue of the risk assessment cannot be approached separately, but always with the connection to other territorial factors, including the planned development [25].

Ensuring the safety of the territory has not only been associated with the spatial planning, but also with the emergency or crisis planning. However, the individual types of plans have been often solved with different regulations and their interconnection has not been optimal. In the final consequence, the effective preparation of the territory has been for possible disasters disrupted. In the future, it will be desirable to create an interconnection between different types of planning documentation [26]. In some cases, the studies also confirm the need for the adjustment in existing legislation, which solve the security of the territory [27].

The establishment of security principles in urban development is also closely linked to their digitization, which can, with appropriate solution, increase the security of citizens [28].

It is therefore clear that the requirements for the safety and sustainability of the territory are closely related to safety planning and the protection of the population [29].

## 2. Materials and Methods

The aim of the article is to characterize a new procedure for determining the requirements of the population protection within the spatial planning process. The requirements for the protection of the population are closely related to the sustainability of the territory and are crucial for its further development. The method was then applied to the conditions of the Czech Republic.

Within the research there has been a search of existing procedures for determining these requirements in the Czech Republic and in selected foreign countries. The obtained information was analyzed, and partial conclusions were drawn using induction and deduction. Using the synthesis of

knowledge, a new, generally applicable procedure for determining the requirements of population protection in spatial planning was subsequently proposed.

The newly created approach presented below keeps the legislative and normative requirements that are, regarding the given issue, valid in the Czech Republic. This approach applies the knowledge to the new principle of their application. For this principle there have been approaches of some foreign countries. Primarily, it is about a division of the application of requirements in a territory without external danger and in a territory with external danger, where the requirements are directed to a specific danger. This fact in the current state of application of the requirements of protection of the population in the Czech Republic is missing, which often causes uncertainties in the application of requirements. From the point of view of the Czech Republic, the procedure is a novelty in its entirety.

### 2.1. Definition of The Term "Population Protection"

The term "population protection" in the Czech Republic means the fulfillment of the civil protection tasks, especially the warning, evacuation, shelter, emergency survival of the population and other measures to ensure the protection of life, health and property [30,31].

One of the strategic goals of the development of population protection in the Czech Republic is to create links between the safety of the population and the sustainability of the territory. This is related to the creation of personnel and material capacities, the education of the population, and the development of safety research [32].

The described definition of population protection is to some extent similar in Switzerland and Germany.

In Switzerland, it has been about the civil protection that sets out the principles for cooperation in dealing with emergencies and crisis situations at the level of the Confederation, the canton, and third parties. It defines the tasks of protection of the population, which include protecting the population and its needs in the event of disasters, emergencies, and armed conflicts, and further contributing to mitigating their consequences and overcoming them [33].

In Germany, the civil protection has been understood to be a necessary range of non-military measures to protect the population, eliminate or mitigate the effects of war on the population, its homes, important businesses, cultural property, and vital workplaces, and to defend important authorities. Civil protection includes, in particular, the self-protection, warning the population, protective buildings, regulation of residence, rescue system, measures to protect health, and measures to protect cultural property [34].

Although the perception of the protection of the population is not entirely the same in different countries, it is always a matter of creating suitable conditions for the population in the event of emergencies of a non-military or military nature. In the following text, the protection of the population will be primarily associated with events of a non-military nature.

### 2.2. Spatial Planning Documentation Structure

Spatial planning in the Czech Republic has been based on spatial analytical documents and spatial planning documentation. Spatial analysis is carried out for the territory of the region and for the administrative territory of the municipality with enlarged jurisdiction. The spatial analytical documents then contain sustainable development analyses of the territory, data about the territory, and other available information or data. The spatial planning documentation consists of the principles of spatial development and the spatial and regulation plan, which are issued in the form of measures of a general nature pursuant to the Administrative Procedure Code [14].

*Principles of territorial development* define the basic principles for territories of the region on their efficient and economic arrangement. They contain the delimitation of areas and corridors of supra-local significance including areas and corridors of the public infrastructure, public utility buildings, public utility measures, buildings, and measures for ensuring state defense and safety.

*The Spatial plan* is processed for the territory of the municipality. The city region development, the protection of its values, the layout of the landscape, and the public infrastructure are set out there. It defines the borders of built-up areas, corridors, areas for public utility buildings, and land reserves.

*The regulatory plan* is processed for a defined area of the land, which is determined by the regional council or the municipal council in the form of measures of a general nature according to the Administrative Procedure Code. It sets the condition for using land, protecting values and the character of the territory, for placing buildings, and for creating a favorable environment in the given area.

### 2.3. Determination of Population Protection Requirements within Spatial Planning

The requirements for the protection of the population within the spatial planning create conditions for ensuring the civil protection, reducing the risk of ecological and natural disasters, and for eliminating their consequences [14].

Currently, the Czech Republic has primarily resolved the following requirements for ensuring [35]:

evacuation of the population and its accommodation,
emergency water supply to the population,
protection against the effects of dangerous substances stored in the territory,
protection against the consequences of a possible terrorist attack on objects the damage of which may cause an emergency situation.

However, the implementation of the population protection requirements is currently problematic in the Czech Republic. The main reason is the ambiguity of the set requirements and thus their difficult real-life application [36].

It is necessary to design a new method for determining population protection requirements within the framework of a spatial plan. The necessity of an amendment is further supported by results from real events and conducted rescue unit exercises [37].

The general scheme of the method for determining the requirements for the protection of the population in the territory is shown in Figure 1.

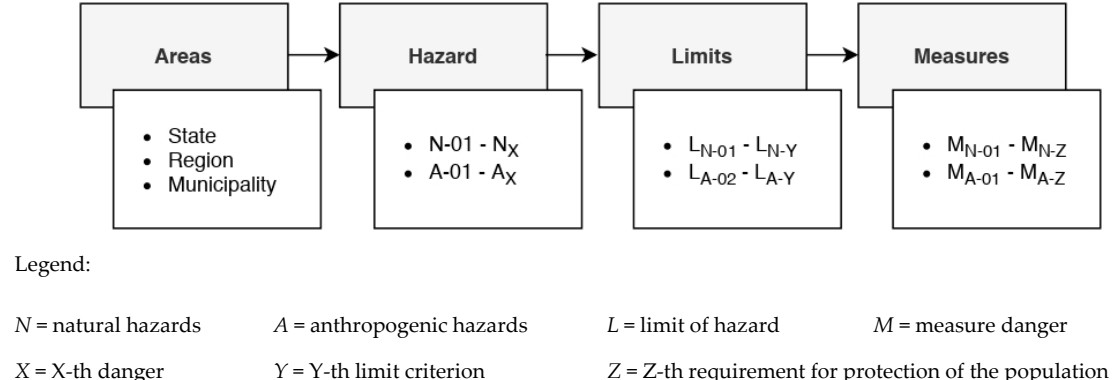

Legend:

| | | | |
|---|---|---|---|
| $N$ = natural hazards | $A$ = anthropogenic hazards | $L$ = limit of hazard | $M$ = measure danger |
| $X$ = X-th danger | $Y$ = Y-th limit criterion | $Z$ = Z-th requirement for protection of the population | |

**Figure 1.** The general scheme of the method for determining the requirements for the protection of the population in the territory.

This general method is the basic framework for determining the requirements for the protection of the population in the area.

A more detailed illustration of the newly proposed method can be seen in Figure 2.

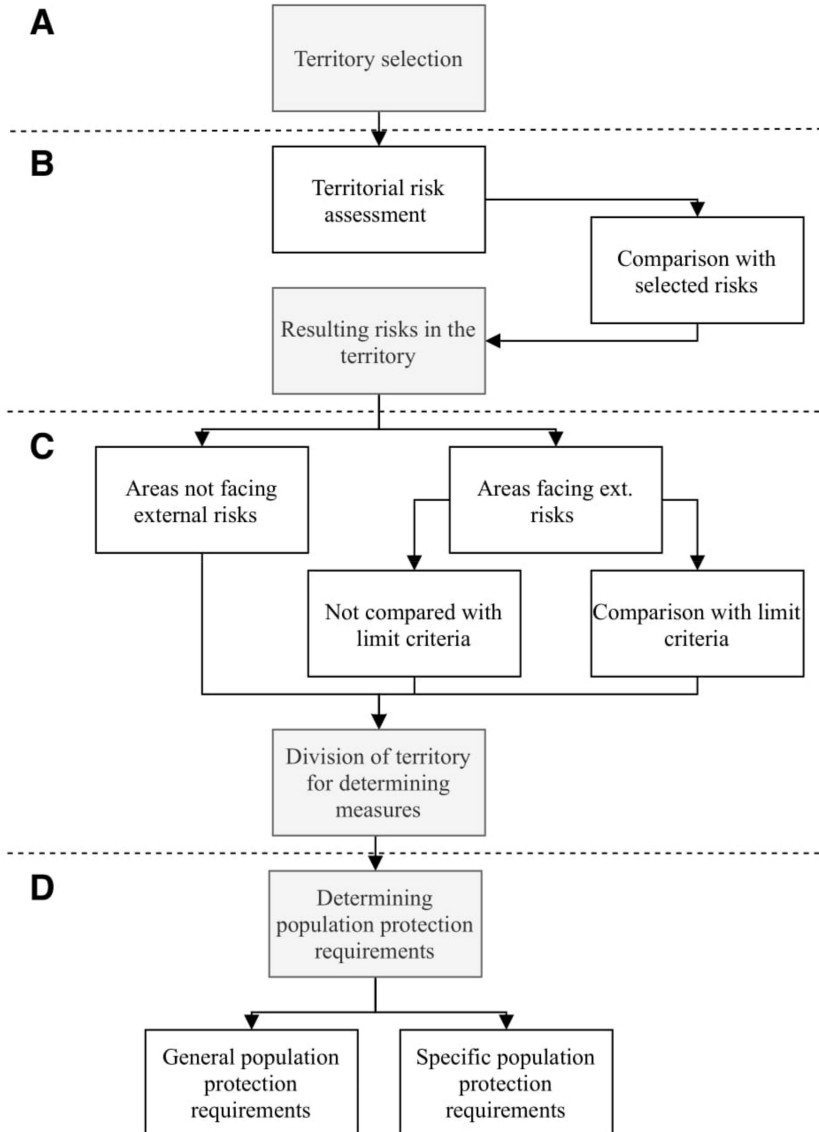

**Figure 2.** A more detailed description of the method for determining the requirements for population protection within spatial planning.

The process of determining the requirements for the protection of the population in a territory can be divided into four steps marked in Figure 2 letters A to D. After the initial step, which is the definition of the territory (step A), possible hazards that may occur in the area are taken into account. The procedure for determining the resulting hazards is described in the following sections of the article. The result of the comparison of the assessed territory with the selected risks is the grouping of territories into "areas not facing external risks" and "areas facing external risks".

Areas facing external risks can be further compared with limit criteria, which set the degree of risk in a given territory in relation to some of the resulting risks. The result is the division of the territory according to limit criteria. The assessor can, however, leave out this step and keep the simple division of areas not facing external risks and areas facing external risks (step C).

For areas not facing external risks, there are further general population protection requirements; for areas facing external risks, there are further specific population protection requirements (step D).

The general requirements of the territory can be described as the basic requirements in terms of the public infrastructure (e.g., access roads, water supply for firefighting) and public utility buildings

(e.g., establishing fire stations). The requirements will also be applied to territories not facing external dangers.

The specific requirements are requirements for a territory due to the existence of external risks (e.g., the occurrence of operations with hazardous chemicals).

The proposed method was further applied for the Czech Republic. For the sake of completeness, it is necessary to state that the general and specific requirements for the protection of the population were determined with regard to the competencies of the Fire and Rescue Service of the Czech Republic, which is the guarantor for the protection of the population in the Czech Republic.

### 2.4. Territorial Risk Assessment

This part of the article deals with the risk evaluation process in the territory, therefore the principle, by which the risks were determined for comparison in the territory in the newly designed method.

The principles defined by technical standards [38,39] can generally be used to assess the hazards and risks of a territory.

The main input for determining the resulting risks in a territory for the Czech Republic have been the analyses already made. These have been mainly the Analysis for the Czech Republic and the subsequent Regional Threat Analysis [40–42]. Within these original works, 72 types of risks have been assessed, which have been divided into three categories. These have been the categories with acceptable hazard, conditionally acceptable hazard, and unacceptable hazard.

The following works determining the resulting hazards in a territory have been aimed only at the level of unacceptable risks. This level contained 22 risks, including floods, extreme wind, radiation accidents, leakage of chemical substances from stationary installations, epidemics, the interruption of the food supply on a large scale, migration, and others. The total list of these risks is given in Table 1. Every risk has been assigned a risk code during selection, grouping the individual hazards into naturogenic (N) and anthropogenic (A), and there have been given the serial numbers.

**Table 1.** The determination of the corrected sum of impacts for individual risks [43].

| Risk Code | Unacceptable Risks | Cor. Sum |
|:---:|:---:|:---:|
| A-02 | radiation accident | 17.0 |
| N-02 | flash floods | 14.0 |
| A-10 | special flood | 14.0 |
| N-01 | natural flood | 13.0 |
| A-01 | leakage of hazardous chemical from stationary equipment | 13.0 |
| A-04 | disruption of large-scale electricity supply | 12.5 |
| A-12 | large-scale breaches of the law | 11.0 |
| A-03 | disruption of large-scale gas supplies | 10.5 |
| A-05 | disruption of the supply of large-scale oil and petroleum products | 10.5 |
| N-04 | extreme prolonged drought | 10.0 |
| A-13 | disruption of the financial and foreign exchange economy of a large-scale scope | 9.5 |
| A-06 | disruption of large-scale drinking water supplies | 9.0 |
| A-11 | large-scale migration waves | 8.5 |
| N-05 | extreme wind | 8.0 |
| N-07 | epidemic | 8.0 |
| N-08 | epizootic | 7.5 |
| A-08 | disruption of the functionality of important electronic communications systems | 7.5 |
| A-09 | disruption of large-scale food supplies | 7.5 |
| N-03 | heavy rainfall | 7.0 |
| N-06 | occurrence of extremely high temperatures | 7.0 |
| N-09 | epiphytis | 5.5 |
| A-07 | breach of information safety of critical information infrastructure | 3.0 |

Unacceptable risks have been then further evaluated in relation to possible consequences [43], namely:

impacts on people's lives and health,
environmental impacts,
economic impacts,
social and social impacts,
international impacts,
impacts on the critical infrastructure.

Subsequently, using the evaluation indices listed in Table 2, a corrected sum of impacts has been determined for individual hazards, characterizing their significance.

**Table 2.** Values for calculating the corrected sum of impacts [43].

| Verbal Evaluation | Evaluation Index | Value for the Corrected Sum of Impacts | |
| --- | --- | --- | --- |
| | | Direct Impacts | Indirect Impacts |
| no impact | 0 | 0 | 0 |
| low impact | 1 | 1 | 0.5 |
| medium impact | 2 | 2 | 1 |
| severe impact | 3 | 3 | 1.5 |

The impacts of individual risk on the given area have been assessed as a part of the evaluation. First there was the assessment of the direct impacts and in the case of low direct impacts (the value of 1 or 0) there was also an evaluation of indirect impacts on the given area. If the value of indirect impacts exceeded the value of direct impacts, this risk has been then assessed as an indirect impact. Direct impacts have been considered those situations, which could endanger the health or lives of people; the indirect impacts were then situations that otherwise interrupt the normal life of the population.

The risks assorted according to their resulting corrected sum of impacts is given in Table 1.

The risks with the highest score should be the subject of priority for the determination of measures from the point of view of protection of the population.

The next phase of the evaluation was focused on the evaluation of the impacts of hazards on the health and life of the population in relation to a certain area [43]. For these needs, it was primarily the territory:

state,
regions,
 municipalities.

The evaluation has been focused on the fact if the given risk has an impact on the evaluated territorial unit (values "yes/no"). The evaluation also reflected that the predominant impacts were direct (red) or indirect (yellow).

The assessment of the effects of the risk on the state level is shown in Figure 3.

The assessment of the impacts of risks on the level of the municipality is shown in Figure 4.

The values of the corrected sums of impacts can be displayed for any territorial unit. It is clear that the significance of the effects of risks varies considerably for different levels of the territory. Within the whole process of the risk assessment in the territory, this is a partial output of the evaluator.

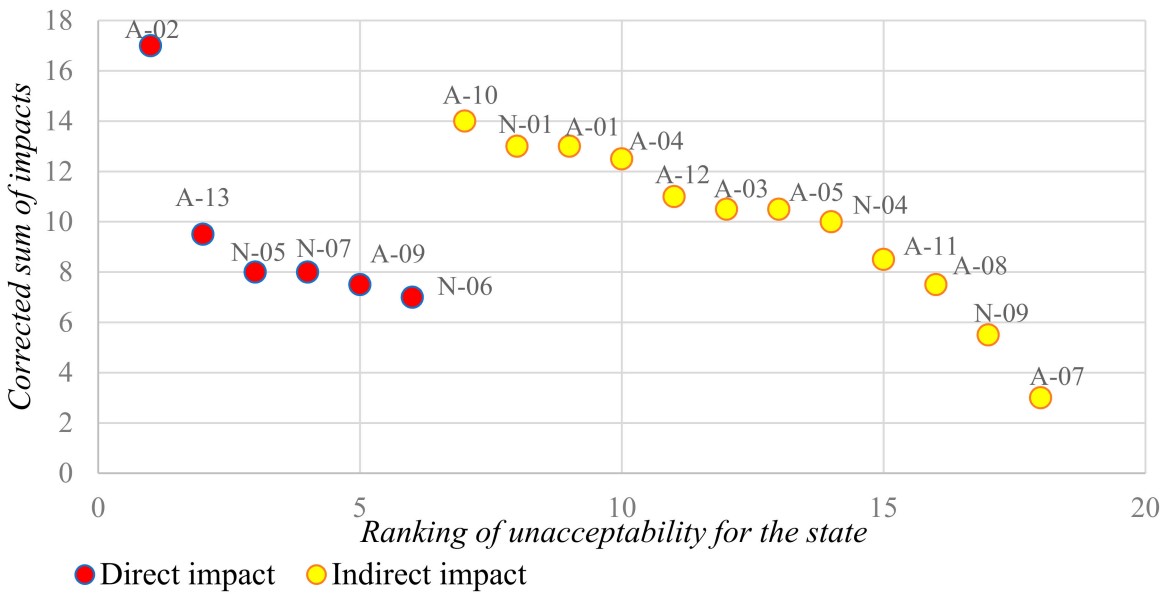

**Figure 3.** State level impact assessment [43].

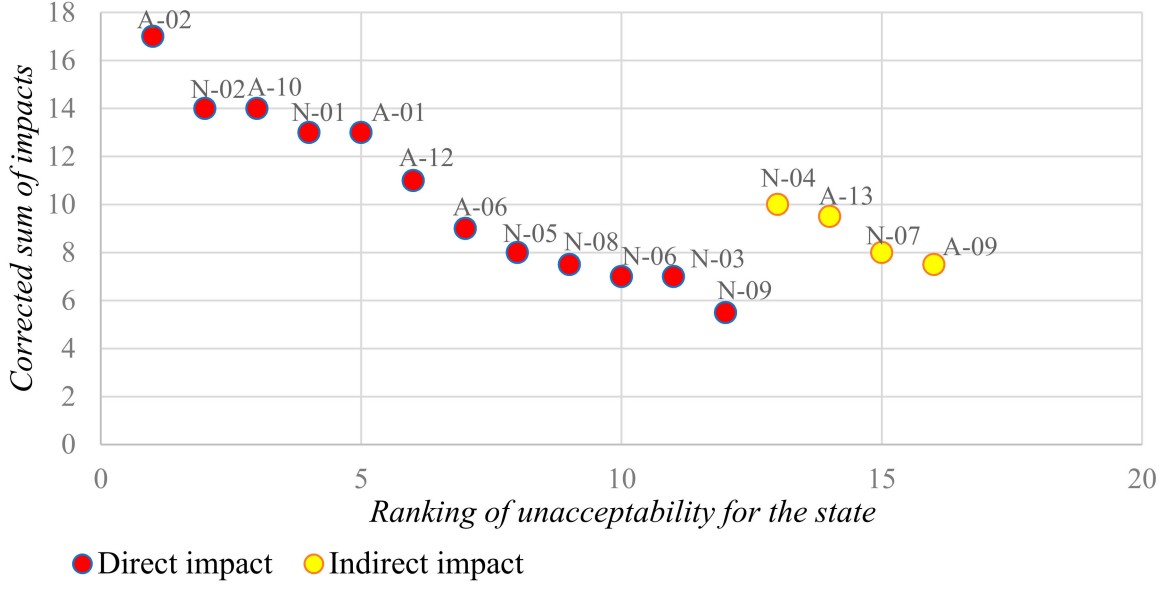

**Figure 4.** Assessment of impacts on the municipality level [43].

*2.5. The Resulting Risks in The Territory*

The partial results obtained within the risk assessment in the area have been the basis for determining the resulting risks. The risks listed in Table 1 have been further assessed from the following points of view [43]:

- The assessment of permanent identifiability in the territory,
- The evaluation of the possibility of preventive measures from the point of view of spatial planning,
- The evaluation of the possibility of preventive measures from the point of view of the competence of the Fire and Rescue Service.

Risks that meet all three above-mentioned aspects have been classified as the resulting risks for subsequent comparison with the assessed area. The resulting risks are shown in Table 3.

**Table 3.** The resulting risks for comparison with the evaluated area [43].

| Risk Code | The Resulting Risks in the Territory |
|-----------|--------------------------------------|
| N-01 | natural flood |
| N-02 | flash floods |
| A-01 | leakage of hazardous chemicals from stationary equipment |
| A-02 | radiation accident |
| A-10 | special flood |

*2.6. Territorial Division for Setting Measures*

The area can be divided further into sub-areas based on the limit criteria (degree of risks of the area) in relation to the resulting hazards in the area.

The possible division of the territory using the limit criteria is given in Table 4.

**Table 4.** Territorial division using limit criteria.

| The Resulting Danger | Limit Criteria for the Division of Risks in the Territory into Sub-Areas |
|----------------------|--------------------------------------------------------------------------|
| territory at risk of natural flooding | - the territory is part of a floodplain<br>- the territory is part of a designated floodplain<br>- territory with a terrain risk factor<br>- the territory does not fall into the above categories (without the given source of danger) |
| territory at risk of flash floods | - territory with a terrain risk factor<br>- the territory does not fall into the above categories (without the given source of danger) |
| territory at risk of leakage of a dangerous chemical substance from a stationary installation | - the territory is part of the emergency planning zone (EPZ) of a building included in group B [44]<br>- the territory is located near a building included in group B, but outside the EZP [44]<br>- the territory is located near a building included in group A [44]<br>- the territory is located near a sub-limit object (i.e., an object with the presence of hazardous substances in a smaller amount than the set limit [44])<br>- the territory is located near hazardous waste repositories<br>- the territory does not fall into the above categories |
| territory at risk of a potential radiation accident | - the territory is part of the EZP of a nuclear facility<br>- the territory is located near the outer edge of the EZP of a nuclear facility<br>- territories that do not fall into the two aforementioned categories, with regard only to measures according to the national radiation emergency plan |
| special flood risk area | - the territory is at risk of special floods<br>- the territory is close to an area at risk of special flooding<br>- the territory does not fall into the above categories (without the given source of threat) |

For each resulting danger, several levels are set, where the first in order represents the worst possible variant and the last level always practically corresponds to the area that is outside the area of interest, i.e., outside the external danger. This scaling can be beneficial for setting priorities when designing specific measures in the given area.

The division of the territory into sub-parts using the limit criteria depends on the professional judgment of the evaluator and may or may not be necessary. In determining the measures to protect the

population of the Czech Republic, this division has been not used. Given the scope of the subsequently proposed measures, the division of the territory seemed redundant.

*2.7. Determination of Population Protection Requirements*

It is possible to process the requirements for the protection of the population for spatial planning based on the previous steps. For the Czech Republic, the population protection requirements have been segmented into the following areas:

- The public infrastructure,
- The public utility buildings,
- The public utility measures (non-constructional).

This division corresponds to the standards for the spatial planning in the Czech Republic. An overview of the requirements for the protection of the population is given in Tables 5 and 6.

**Table 5.** General requirements for population protection in spatial planning.

| Classification of Requirements | | Population Protection Requirements |
|---|---|---|
| public infrastructure | transport infrastructure | General requirements for access roads for the technology of the components of the integrated rescue system in the urban area of the municipality:<br><br>-　　parameters of access roads (width, load capacity)<br>-　　number of access lanes<br>-　　single lane road requirements (vehicle turning)<br>-　　parameters of entrances, passages, and underpasses<br><br>Other requirements for access roads to socially important buildings:<br><br>-　　multidirectional access communications to objects<br>-　　separate lanes in each direction of travel |
| | technical infrastructure | Requirements for water sources for firefighting:<br><br>-　　securing water sources for firefighting<br>-　　location and capacity of such water sources<br>-　　establishment of service stations<br><br>Other requirements for technical infrastructure:<br><br>-　　establishment of temporary repositories for contaminated waste<br>-　　establishing places for the location of mobile water treatment plants |
| | civic amenities | Requirements for fire stations and fire houses:<br><br>-　　establishment of water sources for fire stations<br><br>Other requirements for civic amenities:<br><br>-　　emergency accommodation planning<br>-　　providing warnings and information |
| | public spaces | -　　providing access roads for the components of the integrated rescue system |
| public utility buildings | fire station and fire house | -　　requirements for the location of fire stations and fire houses |

**Table 6.** Specific requirements for population protection in spatial planning.

| Classification of Requirements | | Population Protection Requirements |
|---|---|---|
| public benefit measures | emergency planning zones and risk zones of sub-limit buildings | - providing warnings and information<br>- provision of emergency accommodation<br>- establishment of monitoring of hazardous chemical substances |
| | areas at risk of natural, flash, and special floods | - providing warnings and information<br>- provision of emergency accommodation<br>- ensuring monitoring of water level |

## 3. Discussion and Results

The article presents a new method for determining public protection requirements. This procedure shows Figure 1. The new concept connects the territory with the risks therein, assessing the degree of risk according to delineations of limits and population protection requirements. This given connection is suitable for assessing the territorial safety and it subsequently allows for the determining of general (parts of a territory not facing external dangers) and specific (parts of a territory facing specific dangers) requirements. The novelty of the procedure can be seen in the connection to the identified risks that makes it possible to determine the population protection requirements for any territory when national specifics are considered.

### 3.1. Territorial Risk Assessment

The basis for the risk selection has been the Threat Analysis for the Czech Republic, from which only the unacceptable risks have been chosen. The given analysis also contains acceptable and conditionally acceptable dangers, however their impact on the territory has not been significant and for this reason it has not been selected. By using the results of the already created Threat Analysis for the Czech Republic, the risk evaluation process on the territory was much easier.

Similar Threat analyses have been made in many other countries and their use may be recommended. Every such analysis is always aimed at a given territory and its specific character, which is reflected in the results of the given analyses. An example can be the register of natural disasters reported in the city [45], the results of the safety project in areas with high seismic and environmental risk [46], or the platform for assessing risks in a given territory [47]. The results of the given analyses can then be the basis for the selection of dangers.

The unacceptable risks have been assessed regarding their possible impacts in the territory. Using the assessment indices for individual risks determined a corrected sum of impacts. The danger with the highest value of the corrected sum of impacts regarding the possible consequences in the territory was the subject of interests for the determination of measures in term of the protection of the population. In this part ran the next phase of the evaluation, namely the evaluation of the impacts of the danger on the health and life of the population in relation to a certain territory. This was the territory of the state, region, or municipality.

### 3.2. The Resulting Risks in the Territory

The partial results obtained from the assessment within the evaluation of the danger in the territory, i.e., the risks with the highest value of the corrected sum of impacts in terms of possible consequences in the territory, have been further evaluated from several points of view. These have been the permanent identifiable dangers in the territory, the possibility of preventive measures in terms of spatial planning, and the possibility of preventive measures in terms of the competence of the Fire Rescue Service.

The danger that met all of these aspects was classified as the resulting risks in the territory. Among the resulting dangers were natural and flash floods, leakage of dangerous chemical stationary device, radiological accidents, and specific flood.

At first glance it might seem illogical that the resulting risks do not need to include, for example, a higher risk of drought. The current climate changes that are in effect on our planet steer us towards accounting for such risks [48,49]. However, in the Czech Republic, the preventive measures for these kinds of risks are not in the competency of the fire brigade. For this reason, drought and other similar dangers were not included in the assessment.

It is possible to further divide territories into areas not facing external risks and areas facing external risks based on the resulting risks in the territory.

### 3.3. Territorial Division for Setting mMasures

The territory is then further divided into sub-parts based on limit criteria. If the assessor decides to use this detailed sub-division, there will be the possibility for determining the following requirements regarding the exposure of the territory in terms of danger. If the assessor does not deem it useful to divide the territories into sub-parts, he does not need to perform this step. The sub-division of the territory was not used for the Czech Republic. The reason for this was that the subsequently determined population protection requirements were redundant.

### 3.4. Determination of Population Protection Requirements

The last step was the determination of the general and specific population protection requirements. It is necessary to then implement the determined requirements in legal or technical standards that deal with this issue. For the Czech Republic, the requirements were divided into areas of public infrastructure, public utility buildings, and public utility measures. The given areas are, however, characteristic for the Czech Republic and it is likely that in other countries, there will be a different system of implementation [50,51].

The standards must be amended regarding the individual stages of spatial planning documentation. In principle, it is a matter of elaborating requirements from more general to more specific.

The presented method for determining the requirements for the protection of the population within the spatial planning can be considered as generally applicable. This is evidenced by the successful application of the method in the Czech Republic. In the future, in addition to the direct application of the method, it is also possible to anticipate its development or modification according to the needs of a specific country.

The proposed procedure has been based on the assessment of the territory in terms of the population protection. It takes into account the possible dangers occurring in the territory, it sets out the principles of the territory division in relation to external hazards, and subsequently, in relation to the previous steps, it sets out the requirements for ensuring the safety of the territory. The procedure has not been primarily based on the national regulations of the Czech Republic, but it represents a general, but clearly directed, sequence of solutions to the security of the territory within the spatial planning.

The process of the spatial planning has been used in all countries, although it has certain specific characteristic for each country. Just these specifics can lead to possible modifications to the presented procedure, while it is most likely that the principles of the proposed procedure will be preserved. This versatility has been the benefit of the proposed procedure.

Currently, one of the world's most pressing issues is the Covid19 pandemic. Pandemics are one of the threats identified in the Threat Analysis for the Czech Republic [42]. Although pandemics can trigger a number of measures from the point of view of population protection, such as increased demands on the crisis management system, increased demands on the scope of medical care, or complications in dealing with extraordinary incidents with the occurrence of infected persons, etc., this is not a threat that could be systematically addressed in spatial planning. For this reason, they are also not the subject of the presented new concept of territory protection.

The presented methodology is based on the assessment of risks in relation to the territory. Within the initial analysis of all possible risks were only those risks selected that are by its nature related to the spatial planning or they can be secured by some spatial measures. Therefore, the article does not address the specific risks of Covid19.

## 4. Conclusions

The requirements for ensuring safety and sustainability in a territory are a constant issue in many countries. The given requirements are strongly associated with safety planning and population protection. One of the most significant forms of safety planning is spatial planning.

At present, the requirements for population protection determined in the framework of spatial planning in the Czech Republic are deemed problematic. The reason is the ambiguity and thus also the difficult applicability of the set requirements. The ambiguity of the requirements is mainly due to their considerable obsolescence and at the same time the dynamic development of the discipline of protection of the population.

The article presents "a newly designed method" for determining population protection requirements, which is generally applicable. The procedure is based on a risk assessment in the territory, which is generally considered a necessary input for the creation of preventive measures. For the risk assessment, it is appropriate to use standard procedures presented by global or European standards. Standard procedures can be modified to some extent for the needs of risk assessment in the territory. The important role in the newly proposed procedure play limits of the criteria for the division of the territory in terms of identified risks. The limits of the criteria will be largely related to the safety standards of the country. The results are the measures of population protection determined for areas facing external risks and not facing external risks. The method was then applied to the conditions of the Czech Republic.

The determination of population protection requirements will contribute to increasing the safety and sustainability of a territory. This procedure can be a motivation for the implementation of conceptual changes in the field of spatial planning in the connection with the protection of the population in other countries.

Currently, the application of the proposal within the practical case studies on real territories has been in progress, namely at the guarantor for this area, i.e., at the Fire and Rescue Service. These studies have not been currently closed, i.e., the exact results have not been known. After incorporating the results, the proposal will be formulated and incorporated into the methodological instructions of the Fire and Rescue Service and into the legislation of the Czech Republic.

**Author Contributions:** Conceptualization, J.P., B.M., and S.S.; methodology, J.P., B.M., and S.S.; software, J.P., B.M., and S.S.; validation, L.B. and V.V.; formal analysis, J.P., B.M., and S.S.; investigation, J.P., B.M., and S.S.; resources, J.P., B.M., and S.S.; data curation, J.P., B.M., and S.S.; writing—original draft preparation, J.P., B.M., and S.S.; writing—review and editing, J.P., B.M., S.S., and L.B.; visualization, J.P. and S.S.; supervision, V.V.; project administration, B.M. and S.S.; funding acquisition, J.P. All authors have read and agreed to the published version of the manuscript.

**Funding:** This research was funded by Ministry of the Interior of the Czech Republic, project no. VH20182020042 Population protection in land-use planning and in setting technical conditions for building design.

**Acknowledgments:** The authors would like to thank the representatives of the Ministry of the Interior of the Czech Republic and the Institute for the Protection of the Population for their cooperation.

**Conflicts of Interest:** The authors declare no conflict of interest.

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
