# Peer review of "Planning of Safety of Cities and Territory from the Point of View of Population Protection in the Czech Republic"

_sustainability, doi:10.3390/su12229487_

Round 1

Author Response

Dear reviewer,

authors of the article thank you for all the comments and in the following text there are annotated modifications of the text to the article, which were made on the basis of your comments.

Reminder:

 -From a theoretical point of view, we consider there is not a solid theoretical background that conveniently defines the object of study. These concepts are defined vaguely. This section should answer key questions such as: How does civil protection or territorial security relate to key European Union concepts such as territorial cohesion or territorial sustainability?

It is recommended that bibliographic references be used on the concept of territorial cohesion, which ultimately aims to optimize the conditions of security and protection for society. That is, the more cohesion, the lower the risks to the population. There is certainly a need for reflection on this, introducing new bibliographic references, and not brief descriptions of what these concepts entail in Germany or Switzerland. In this sense, the bibliographic resources used by the article are very scarce.

Response:

The relation between civil protection and the territorial security were in the previous version of the article mainly related to the European Union linked to „A shared vision, a common approach: a stronger Europe. Global strategy of foreign and safety policy of the European Union, 2016 [1]“, to „Union's general action program on the environment up to 2020 “Living well within the limits of our planet, 2013 [2]“ and to „Review of the EU Sustainable Development Strategy (EU SDS) - Renewed Strategy, 2006 [3]“.

New links are added to the prevent the occurrence and consequences of extraordinary incidents, crisis situations and disasters, which are based on the assessing the risks of the territory and determining preparedness for dealing with disasters. It is a fixed part of the European Union's security principles, which are a part of the so-called civil protection mechanisms. See literature sources [5,6].

There was also added a link to the global concepts of a risk assessment and response to them, which creates a global strategic background for addressing this issue from 1994 to the present. See literature sources [7-10].

The new documents require from Member States to work intensively on the risk assessment and to prepare documents for a disaster response. One of the documents are also the conditions set within the spatial planning and it is a direct connection to the presented new concept of protection of the territory in terms of protection of the population.

With the combination of original and new documents comes to the necessary document for the linking the issue of population protection in the Czech Republic with the mechanisms of the European Union.

References to the solution of the issue of spatial planning and its connection with civil protection in Germany and Switzerland are additional information. These countries are presented here due to the similarity of spatial planning systems.

Reminder:

 - From the point of view of the content structure, in the article does not appear a results section. These appear without differentiating in the methodological section. It is recommended to discuss them in a separate chapter or in the discussion section, calling this section "discussion and results".

Response:

The Discussion section has been renamed and extended to Discussion and results. This section has been added of the conclusions of subchapters, which present the individual steps of the proposed procedure for determining the requirements for the protection of the population in the territory. These subchapters correspond to the steps described in the paper.

Reminder:

 -It is recommended to extend the conclusions.

Response:

The conclusions in the article have been extended.

Reminder:

- The references in the text do not follow the journal rules. The authors use the APA System, not numbers.

 Response:

The reference and links in the text have been in the translated article modified according to the rules of a diary. These changes are not recorded by the Change tracking function.

Reminder:

 -It would be very opportune to comment briefly on the possible interrelationships between population protection and the measures being taken to stem the spread of COVID-19.

Response:

To the Discussion and results part has been the additional text inserted (see below).

„ Currently, one of the world's most pressing issues is the Covid19 pandemic. Pandemics are one of the threats identified in the Threat Analysis for the Czech Republic [33]. Although pandemics can trigger a number of measures from the point of view of population protection, such as increased demands on the crisis management system, increased demands on the scope of medical care or complications in dealing with extraordinary incidents with the occurrence of infected persons, etc., this is not a threat that could be systematically addressed in spatial planning. For this reason, there are also not the subject of the presented new concept of territory protection. “

The presented methodology is based on the assessment of risks in relation to the territory. Within the initial analysis of all possible risks, were only those risks selected that are by its nature related to the spatial planning or they can be secured by some spatial measures. Therefore, the article does not address the specific risks of COVID-19.

In conclusion, the authors would like to thank you once again for your time in reading and commenting on the article.

Sincerely

Authors

Reviewer 2 Report

The work seems not very thorough, with a limited and little updated bibliography

Author Response

Dear reviewer,

authors of the article thank you for all the comments and in the following text there are annotated modifications of the text to the article, which were made on the basis of your comments.

Reminder:

The work seems not very thorough, with a limited and little updated bibliography.

Response:

The authors worked in the text with literary sources that are related to the development of requirements to ensure the protection of the population and sustainability in the Czech Republic and abroad. They could not completely avoid the literary sources of an older date.

However, these are mainly the laws or concepts of the Czech Republic that define the area. This obsolescence only supports the fact that the authors point out in the abstract of the article: “The existing population protection requirements contained in the spatial plan have been unsatisfactory for a long time. The main issue is that they are out of date and difficult to apply.”

In the modified article, the literary sources have been extended to include global concepts for assessment of the risks and responses to them, and European Union strategy papers related to civil protection mechanisms.

This led to the further extension of literary sources and in connection with the previous documents should provide comprehensive ties by the protection of the population, planning and sustainability of a territory.

In conclusion, the authors would like to thank you once again for your time in reading and commenting on the article.

Sincerely

Authors

Reviewer 3 Report

The paper deals with an important issue of population protection in spatial planning. However, the article has some weaknesses.

  1. While reading the article, I got the impression that all the results come mainly from one source- Pokorny et al. (2019). It was cited 8 times. After the analysis of the references it occurred that these were the results of a project in which 3 from 5 of the authors of the article took part. In my opinion, it should be stated what part of the methodology comes from the project and what is a novelty.
  2. Section 2, line 81: There is no information what has been taken from existing procedures for the Czech Republic and analyzed countries. It should be specified what is new in the procedure.
  3. How the values given in Figure 1 were established?
  4. The title of the paper should be changed because of the repetition of the word "protection". It could be sound, for example: Planning of protection of cities and territory - a case study of the Czech Republic?
  5. There are also some repetitions of words and punctuation errors:
    • line 32-33: repetition of the phrase "Territorial safety and population protection...";
    • line 45-46: repetition of the phrase "in the Czech Republic";
    • line 175: lack of dot;
    • line 199 refers to table 2 at first, line 209 refers to table 1, so in my opinion, the tables should be replaced (renumbered);
    • line 227: lack of comma;
    • line 229: lack of dot.

Author Response

Dear reviewer,

authors of the article thank you for all the comments and in the following text there are annotated modifications of the text to the article, which were made on the basis of your comments.

Reminder 1:

While reading the article, I got the impression that all the results come mainly from one source- Pokorny et al. (2019). It was cited 8 times. After the analysis of the references it occurred that these were the results of a project in which 3 from 5 of the authors of the article took part. In my opinion, it should be stated what part of the methodology comes from the project and what is a novelty.

Response:

The basis of the newly proposed concept of territory protection in terms of population protection mentioned in the article is based on the Security Research of the Czech Republic. One of the outputs of the project is also "Summary research report of the project Protection of the population in spatial planning and in determining the technical conditions for the design of buildings. Procedure for defining population protection requirements in spatial planning. “ However, this is not the only one output. Another essential part of the project was focused on the buildings.

The area of spatial planning was within the project guaranteed by 3 authors of the article. In the cooperation with the other two authors of the article, there came to the modification of specification at the limit criteria for territorial division (see Table 4) and the resulting requirements for territory (see Table 5 and Table 6). The reason was the experience gained with the practical application of the originally set limit criteria and requirements, in which the other two authors of the article participated. The practical verification was carried out in cooperation with the Fire and Rescue Service of the Moravian-Silesian Region for the territory of the Moravian-Silesian Region.  

Reminder 2:

Section 2, line 81: There is no information what has been taken from existing procedures for the Czech Republic and analyzed countries. It should be specified what is new in the procedure.

Response:

The third paragraph was added into the chapter Materials and Methods.

Reminder 3:

How the values given in Figure 1 were established?

Response:

Figure 1 shows the general scheme of the method for determining the requirements for the protection of the population in the territory. This scheme is presented in general, i.e. without specific values of a certain territory. For this reason, the values are expressed non-numerically, i.e. using the variables x, y, z, which mean an indefinite number. For example, one territory can have several natural risks, the total number of them is generally expressed by the variable x. The first natural risk is marked as N-01, the second natural risk is marked as N-02 and the last natural risk is marked as N-x. The same principle expresses the number of limits by the variable y and the number of measures by the variable z. Therefore, Figure 1 does not represent specific values, but it generally outlines the basic principle.

Reminder 4:

The title of the paper should be changed because of the repetition of the word "protection". It could be sound, for example: Planning of protection of cities and territory - a case study of the Czech Republic?

Response:

The title of the article was changed to:

Planning of safety of cities and territory from the point of view of population protection in the Czech Republic

Reminder 5:

There are also some repetitions of words and punctuation errors:

  • line 32-33: repetition of the phrase "Territorial safety and population protection...";

Response: Repetition of the phrase removed

  • line 45-46: repetition of the phrase "in the Czech Republic";

Response: Repetition of the phrase removed

  • line 175: lack of dot;

Response: A dot added

  • line 199 refers to table 2 at first, line 209 refers to table 1, so in my opinion, the tables should be replaced (renumbered);

Response:

Line 199 refers to other risks from the complete list of 22 risks. However, more relevant information is given in Table 1, which presents the values for calculating the corrected sum of impacts. The results of these values are then given in Table 2, which also shows the complete list of risks, which refers to the line 199 (now line 230). For this reason, it is not appropriate to renumber the tables.

  • line 227: lack of comma;

Response: A comma added

  • line 229: lack of dot.

Response: A dot added

In conclusion, the authors would like to thank you once again for your time in reading and commenting on the article.

Sincerely

Authors

Round 2

Reviewer 1 Report

All the comments have been revised. As far I am concern, the paper can be published.

Author Response

The authors would like to thank you once again for your reading time and commenting the article.

Sincerely,

Authors

Reviewer 2 Report

The theme is interesting. We suggest the following changes:
- better express how this approach can be replicated in other countries (even outside the EU)
- improve the scientific bibliography, especially recent papers
- give greater importance to the practical part with the case study

Could it be possible to focus on the risk associated with the covid emergency?

Author Response

Dear reviewer,

authors of the article thank you for all the comments and in the following text there are annotated modifications of the text to the article, which were made on the basis of your comments.

Reminder:

The theme is interesting. We suggest the following changes:

- better express how this approach can be replicated in other countries (even outside the EU)

Response:

The authors added into the Discussion and Results part information of the possible use of the presented process in other countries (see lines 423–433).

Reminder:

- improve the scientific bibliography, especially recent papers

Response:

The scientific bibliography has been expanded in the Introduction part of additional contributions, which have been published in the latest period.  

Reminder:

- give greater importance to the practical part with the case study

Response:

Within the ending of the article has been introduced the current state of the practical application of the proposed procedure (see lines 469–473).

Reminder:

Could it be possible to focus on the risk associated with the covid emergency?

Response:

At previous review has been into the Discussion and results part the additional text inserted (see below).

„ Currently, one of the world's most pressing issues is the Covid19 pandemic. Pandemics are one of the threats identified in the Threat Analysis for the Czech Republic [42]. Although pandemics can trigger a number of measures from the point of view of population protection, such as increased demands on the crisis management system, increased demands on the scope of medical care or complications in dealing with extraordinary incidents with the occurrence of infected persons, etc., this is not a threat that could be systematically addressed in spatial planning. For this reason, there are also not the subject of the presented new concept of territory protection. “(see lines 434–440)

Newly the part has been expanded, which concerns the Covid 19 in the Discussion and results part (see lines 441–444)

In conclusion, the authors would like to thank you once again for your reading time and commenting the article.

Sincerely,

Authors

Reviewer 3 Report

The paper was improved according to my comments. However, I do not agree with the explanation concerning replacing table 1 and table 2. In my opinion, the paper should cite the tables in chronological sequence. So the study should be designed appropriately.

Moreover, in my opinion, the English style requires some modifications by a translator or native speaker.

Author Response

Dear reviewer,

authors of the article thank you for all the comments and in the following text there are annotated modifications of the text to the article, which were made on the basis of your comments.

Reminder:

The paper was improved according to my comments. However, I do not agree with the explanation concerning replacing table 1 and table 2. In my opinion, the paper should cite the tables in chronological sequence. So the study should be designed appropriately.

Response:

The order of tables 1 and 2 has been changed so that the references in the text correspond to the order of the tables.

Reminder:

Moreover, in my opinion, the English style requires some modifications by a translator or native speaker.

Response:

The English used in the article has been repeatedly checked and edited by a native speaker.

In conclusion, the authors would like to thank you once again for your reading time and commenting the article.

Sincerely,

Authors

Round 3

Reviewer 2 Report

accepted in this form